# Interactomic affinity profiling by holdup assay: Acetylation and distal residues impact the PDZome-binding specificity of PTEN phosphatase

**Pau Jané**[1], **Gergő Gógl**[1], **Camille Kostmann**[1], **Goran Bich**[1], **Virginie Girault**[2¤], **Célia Caillet-Saguy**[2], **Pascal Eberling**[1], **Renaud Vincentelli**[3], **Nicolas Wolff**[2], **Gilles Travé**[1]*, **Yves Nominé**[1]*

**1** (Equipe labelisée Ligue, 2015) Department of Integrative Structural Biology, Institut de Génétique et de Biologie Moléculaire et Cellulaire (IGBMC), INSERM U1258/CNRS UMR 7104/Université de Strasbourg, Illkirch, France, **2** Unité Récepteurs-canaux, Institut Pasteur, UMR 3571/CNRS, Paris, France, **3** Architecture et Fonction des Macromolécules Biologiques (AFMB), CNRS/Aix-Marseille Université, Marseille, France

¤ Current address: Technical University of Munich, School of Medecine, Institute of Virology, Munich, Germany
* traveg@igbmc.fr (GT); nominey@igbmc.fr (YN)

**Data Availability Statement:** All relevant data are within the manuscript and its Supporting information files.

## Abstract

Protein domains often recognize short linear protein motifs composed of a core conserved consensus sequence surrounded by less critical, modulatory positions. PTEN, a lipid phosphatase involved in phosphatidylinositol 3-kinase (PI3K) pathway, contains such a short motif located at the extreme C-terminus capable to recognize PDZ domains. It has been shown that the acetylation of this motif could modulate the interaction with several PDZ domains. Here we used an accurate experimental approach combining high-throughput holdup chromatographic assay and competitive fluorescence polarization technique to measure quantitative binding affinity profiles of the PDZ domain-binding motif (PBM) of PTEN. We substantially extended the previous knowledge towards the 266 known human PDZ domains, generating the full PDZome-binding profile of the PTEN PBM. We confirmed that inclusion of N-terminal flanking residues, acetylation or mutation of a lysine at a modulatory position significantly altered the PDZome-binding profile. A numerical specificity index is also introduced as an attempt to quantify the specificity of a given PBM over the complete PDZome. Our results highlight the impact of modulatory residues and post-translational modifications on PBM interactomes and their specificity.

## Introduction

PDZs, named from the three proteins PSD-95, DlgA and ZO1, are globular protein domains that adopt a conserved antiparallel β-barrel fold comprising 5 to 6 β-strands and 1 to 2 α-helices. PDZ domains are involved in diverse cellular activities, such as cell junction regulation,

**Funding:** Acknowledgements Institutional support from: Centre National de la Recherche Scientifique (CNRS), Université de Strasbourg, Institut National de la Santé et de la Recherche Médicale (INSERM), Région Alsace. "Post-doctorants en France" program of the ARC foundation (G.G.). European Union´s Horizon 2020 research and innovation program under the Marie Sklodowska-Curie (grant agreement No 675341) (G.T.). Ligue contre le cancer (équipe labellisée 2015) (G.T.). National Institutes of Health (Grant R01CA134737) (G.T.). Canceropôle Grand-Est (projet Emergent). French Infrastructure for Integrated Structural Biology (FRISBI). The funding sources are not involved in data collection, analysis and interpretation; they did not participate to the manuscript writing nor in the decision to submit the article.

**Competing interests:** The authors have declared that no competing interests exist.

**Abbreviations:** BI, Binding Index; FP, Fluorescence Polarization; HPV, Human Papilloma Virus; MBP, Maltose-Binding Protein; PBM, PDZ-binding motif; PDZ, PSD95/DLG/ZO-1; PTEN, phosphatase and tensin homolog deleted on chromosome 10; PTM, Post-Translational Modifications; TRX, Thioredoxin.

cell polarity maintenance or cell survival [1]. PDZs recognize short linear motifs (called PDZ Binding Motif or PBMs) that follow particular sequence requirements and are mostly located at the extreme carboxyl terminus of target proteins [2]. The human proteome contains 266 PDZ domains dispersed over 152 proteins [3] and thousands of presumably disordered C-termini matching a PBM consensus [4].

The core of a C-terminal PBM is formed by four residues, which are disordered in the unbound state but form, upon binding, an anti-parallel β-strand that inserts between a β-strand and a α-helix of the PDZ domain. A C-terminal PBM contains two conserved residues (positions are thereafter numbered backwards from the C-terminus, starting at p-0): a hydrophobic residue at p-0 and a characteristic residue at p-2, which actually determines the PBM classification: Ser/Thr for class I, a hydrophobic residue for class II and Asp/Glu for class III (Fig 1A). Other positions located within or upstream of the core motif may also modulate the binding affinity ([5–8] and reviewed in [3]). In particular, systematic mutagenesis experiments have shown that amino acid replacements at positions -1, -3, -4 and -5, and sometimes even at upstream positions, can strongly alter the binding properties depending on the PDZ domain [9–12]. We and others have also shown that the length of the peptides or the upstream or downstream sequences of the PDZ constructs used may influence the binding affinity in the assays [13–17].

Additionally, post translational modifications (PTM) at residues within or upstream of the PBM core are susceptible to alter the binding affinity for PDZ [18], and therefore the network involving PDZ/PBM recognition. Protein acetylation is an example of PTM, that mainly targets lysine residues. Acetyltransferases catalyze the transfer of an acetyl group from acetyl-coenzyme A to the ε-amino group of a lysine residue, inducing the neutralization of the positive charge of the lysine side chain. The reaction can be reversed by lysine deacetylases. By modifying the chemical nature of the protein, the acetylation process may alter its binding properties. In particular, an acetylated protein may become "readable" by specialized acetyl-lysine binding domains such as bromodomains [19]. Acetylation occurs in a large variety of

**A)**

| Position | -12 | -10 | -8 | -6 | -4 | -3 | -2 | -1 | 0 |
|---|---|---|---|---|---|---|---|---|---|
| Class I | | | | | | X | [S/T] | X | [φ] |
| Class II | | | | | | X | [φ] | X | [φ] |
| Class III | | | | | | X | [D/E] | X | [φ] |

**B)**

| Position | -12 | -11 | -10 | -9 | -8 | -7 | -6 | -5 | -4 | -3 | -2 | -1 | 0 |
|---|---|---|---|---|---|---|---|---|---|---|---|---|---|
| PTEN_11 | | | D | E | D | Q | H | T | Q | I | T | K | V |
| PTEN_Ac | | | D | E | D | Q | H | T | Q | I | T | *ac*K | V |
| PTEN_KR | | | D | E | D | Q | H | T | Q | I | T | R | V |
| PTEN_13 | P | F | D | E | D | Q | H | T | Q | I | T | K | V |

**Fig 1. Summary of the PBM classes and the sequences studied in this work.** **(A)** A classification has been proposed for the PDZ domains according to the sequence consensus observed for the bound PBMs, and is shown here. See text for details. **(B)** The sequences of the three 11-mer and the 13-mer PBM peptides used in this work are represented. The 11-mer and 13-mer wild-type sequences correspond to the PDZ binding motif of the canonical isoform I of human PTEN (Uniprot ID: P60484), encompassing residues 393–403 or 391–403, respectively. The mutated residues as compared to wild-type are indicated in bold in peptide sequences. *ac*K: acetylated lysine.

protein substrates and plays important roles in protein regulation, DNA recognition, protein/protein interaction and protein stability [20]. Originally widely described for histone proteins, it has also been observed for a growing number of non-histone proteins [21], such as PTEN [22].

PTEN (phosphatase and tensin homolog deleted on chromosome 10) is a lipid phosphatase protein located in the cell nucleus with a prominent tumor suppressor activity. When brought to the plasma membrane, PTEN is able to antagonize the phosphatidylinositol 3-kinase (PI3K), inhibiting the PI3K-dependent cell growth, survival and proliferation signaling pathways [23]. Interestingly, PTEN harbors a class I PBM–ITKV$_{COOH}$–that appears to be critical for regulating its functions [24–27]. The PDZ binding to the PTEN PBM leads to a stabilization of PTEN and an increase of its catalytic activity [28]. The PBM of PTEN presents several original characteristics. On the one hand, a structural study performed by NMR, a method highly sensitive to aggregation, revealed an unconventional mode of binding of PTEN to the unique PDZ domain of the human kinase MAST2 [29]: while the core of the PTEN PBM displays a canonical interaction with the PDZ domain, a Phe residue at p-11 (F392) distal from the core PBM establishes additional contacts with MAST2 through a hydrophobic exosite outlined by the β2- and β3-strands of the PDZ domain. On the other hand, lysine K402, located at the p-1 position of the PBM core in PTEN, has been suggested to represent a putative target of an acetylation reaction that might modulate PTEN binding to PDZ domains and thereby affects other PTEN activities [22]. Remarkably, those original characteristics of the PBM of PTEN (unconventional PDZ binding mode of PTEN and potential modulation by acetylation) have been examined only in context of interaction with a limited number of PDZ domains. It is thereafter interesting to cover their impact on the interactome with the full family of known human PDZ domains (the PDZome), thus requiring the use of a high-throughput screening method, as the holdup.

The holdup method is a chromatographic approach in solution developed in our group that allows to measure the binding strength of a peptide, attached to a resin, against a library of domains of a same family. We initially proposed this method to explore the interaction between PBM peptides and the human PDZ domains [30]. Briefly, a soluble cell lysate containing individually overexpressed PDZ domain is incubated until equilibrium with a calibrated amount of streptavidin-resin saturated either with the target biotinylated PBM peptide or with biotin as a reference. The flow-throughs containing the unbound protein fraction are recovered by filtration and loaded on a capillary electrophoresis instrument to quantify the amount of remaining free PDZ. The stronger the steady-state depletion of the PDZ domain in the flow-through as compared to the reference, the stronger the PDZ/PBM binding interaction. The assay is particularly suited to quantitatively evaluate and compare large numbers of interactions. This method delivers, for each PBM/PDZ pair, a "binding intensity" (BI), whose value can in principle range from 0.00 (no binding event detected) to 1.00 (maximal binding event). The approach has been automated [31] and the human PDZ library was recently extended to the complete 266 PDZ domains known in human proteome [32]. The full processing leads to a binding profile, i.e. a list of binding strengths in decreasing order exhibited by a given PBM towards the entire PDZome. The high accuracy and efficiency of the holdup assay have been validated previously [4,16,31,33]. Very recently, a manual version of the holdup assay with purified samples and using widespread benchtop equipment has been implemented and has proven to be reliable [34].

In the present work, we investigated how the acetylation at position K402 in PTEN (–IT$^{Ac}$KV$_{COOH}$–thereafter corresponding to p-1 position in the PBM), would alter the binding affinity profile of the PTEN C-terminus to the full the PDZome. We also assessed the contribution of the K402R mutation expected to preserve the positive charge and the overall bulkiness

of the lysine residue, as well as the effect of the inclusion of two residues in N-termini, encompassing the exosite previously described and including the p-11 hydrophobic phenylalanine (F392). For these purposes, we combined the updated high-throughput holdup assay with competitive fluorescence polarization (FP) measurements allowing to convert each BI value into affinity. We obtained all the affinities of the complete human PDZ library for wild-type, acetylated and mutated versions of the PBM of an 11-mer PTEN C-terminal peptide as well as an extended 13-mer peptide. We also introduced an attempted "specificity index"–or conversely a "promiscuity index"–to quantify the PDZome-binding specificity of each peptide. The results show that acetylation affects the affinities for the PDZome and highlight the importance of the exosite in modulating the PDZome specificity for the PDZ-binding motif of PTEN.

## Material and methods

### Protein expression and purification

The 266 PDZ domains that constitute the PDZ library ("PDZome V.2") used in the present work were produced using constructs with optimized boundaries as described previously [35]. All the genes were fused to Maltose-Binding Protein (MBP) or Thioredoxin (THR) tag often used to facilitate the solubilisation of the over-expressed protein, by cloning them into the pETG41A or pETG20A plasmid, respectively. The expressions in *E.coli* resulted in a recombinant protein fused to an N-terminal solubility tag (His-MBP or His-TRX). The expressed tag-PDZ concentrations were quantified using capillary gel electrophoresis and cell lysates were diluted to reach approximately 5 μM tag-PDZ before freezing in 96-well plates. A detailed protocol of the PDZ library production, expression and benchmarking can be found in [32]. PDZ domains are named according to their originating protein name followed by the PDZ number (e.g. NHERF1-1 as the first PDZ domain of the NHERF1 protein).

For FP assay, tandem affinity purified $His_6$-MBP-PDZ proteins were used. Cell lysates were purified on Ni-IDA columns, followed by an MBP-affinity purification step. Protein concentrations were determined by far-UV absorption spectroscopy. A detailed protocol has been published previously [4].

### Peptide synthesis

All 11-mer biotinylated peptides (PTEN_11, PTEN_Ac and PTEN_KR) were chemically synthesized on an ABI 443A synthesizer with Fmoc strategy by the Chemical Peptide Synthesis Service of the IGBMC, while the 13-mer PTEN_13 peptide was purchased from JPT Innovative Peptide Solutions with 70%–80% purity (Fig 1B). A biotin group was systematically attached to the N-terminal extremity of those peptides via a TTDS linker. For FP assays, fluorescent peptides were prepared by directly coupling fluorescein to the N-terminus (fRSK1: fluorescein-KLPSTTL; fpRSK1: fluorescein-KLPpSTTL; f16E6: fluorescein-RTRRETQL). Predicted peptide masses were confirmed by mass spectrometry. Due to the lack of aromatic residue, peptide concentrations were first estimated based on the dry mass of the peptide powders and subsequently adjusted by far-UV absorption (at 205 and 214 nm) [36,37].

### Holdup assay

The holdup assay was performed on a Tecan freedom Evo200 robot with 384-well plates in singlicate for the three 11-mer PTEN variants and the 13-mer PTEN peptide as described in [31,32]. Briefly, prior to interaction assay, 2.5 μL of streptavidin resin was saturated in each well with 20 μL of biotinylated PBM peptides (42 μM) and then washed twice with an excess

of free biotin, while the reference resin was incubated only with biotin. Right before the holdup experiment, the PDZ library was spiked with an internal standard of lysozyme. Then, the biotin- or PBM-saturated resin was incubated, each in a distinct well of a 384-well plate, with complete cell lysates diluted so that the concentration of the tag-PDZ present in the crude extract is adjusted at 4 μM. After a sufficient time to reach complex equilibrium (15 min.), a fast and mild filtration step is performed and the tag-PDZ concentrations were measured by capillary electrophoresis instrument (LabChip GXII, PerkinElmer, Massachusets, USA). Standard markers were used to convert migration time into molecular weight on the LabChip software and inappropriate molecular weight markers were corrected or excluded.

## Holdup data quality check and processing

Holdup data can be missing for some tested pairs mainly for three reasons: i/biochemical issues, specially when the overexpressed domain is not concentrated enough in the sample, ii/ acquisition problems mainly because of a misreading of the Caliper data, iii/technical difficulties related to data processing. For points i/ and ii/, many efforts have been made to optimize the expression and to carefully run the LabChip GXII instrument in the best conditions. For point iii/, we developed bioinformatics processing tools in order to improve the accuracy and reproducibility of the intensity measurement of the tag-PDZ peak in the chromatogram [38]. Briefly a baseline correction of the electropherograms is first performed in order to remove the background noise and extract the real intensities using Python package available in https:// spikedoc.bitbucket.io under the name of SPIKE.py [39,40]. Then intensities are normalized using the internal standard (lysozyme as previously mentioned) to correct potential variations over all the protein concentrations. Lastly, both the sample and the reference electropherograms were superimposed by adjusting the molecular weight on the X-axis according to a linear transformation (translation and dilation) of the sample electropherogram as compared to the reference one.

Beyond the purpose of this article, we have accumulated several tens of thousands of PDZ/ PBM interaction data with the holdup protocol used here, most of them being replicated. Based on this large dataset and prior to the present study, experienced holdup data curators have established four quantitatively evaluable quality criteria to be combined, in order to retain or discard data during visual inspection. Individual electropherograms must display a sufficiently high intensity of the normalization peak (criterion 1) and of the tag-PDZ peak (criterion 2) while the signal of crude extract should be kept as low as possible comparatively to the tag-PDZ peak (criterion 3) (Fig 2A). When superimposing the two reference and PBM electropherograms, the elution profiles must be sufficiently aligned (criterion 4) (Fig 2B). In order to rationalize and accelerate data curation, we assigned to each criterion an individual quality score ranging from 0 to 1 from the lowest to the highest quality data (Fig 2C). To avoid a cut-off effect, a linear or quadratic transition was introduced depending on the quality criteria type. The product of the resulting individual scores led to a global quality score in the 0-to-1 range. We calculated such global scores for replicated holdup data sets recorded previously, then compared the scores of the data that had been either rejected or retained based on reproducibility. This allowed us to semi-empirically set a threshold value of 0.6 for the global score, which maximizes the true positive rate and minimizes the false negative rate. This threshold was automatically used to distinguish data to be rejected from those to be retained in a way that generally agrees with rejections done according to reproducibility. This semi-automatic processing has been used for the datasets presented in this study recorded in singlicate, leading to a percentage of rejection never exceeding 10%.

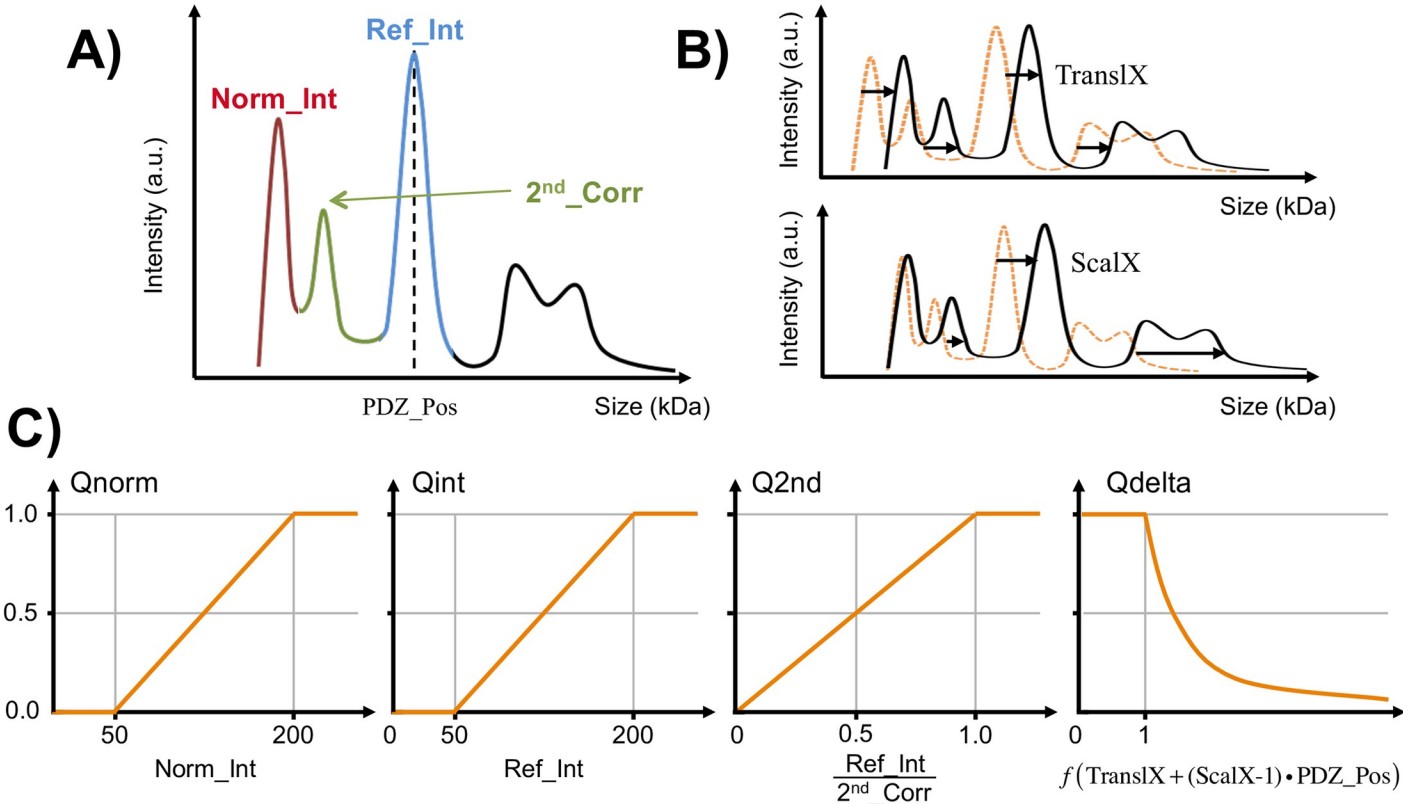

**Fig 2. Quality criteria and their conversion to the individual quality scores used to filter the holdup data.** (**A**) A schematized electropherogram showing intensities of the normalization peak (*Norm_Int*) and of the MBP-PDZ peak (*Ref_Int*) visible in the red and blue regions, respectively. The region in green corresponds to the proteins of the crude extract, which is supposed to be kept low as compared to *Norm_Int* and *Ref_Int* in order to ensure that the MBP-PDZ is not under-expressed (**B**) The linear transformation used to superimpose the sample and reference electropherograms should be as neutral as possible: The *TranslX* translation factor and the *ScalX* scaling coefficient (>1 for dilation or <1 for a contraction) should be as close as possible to 0.0 and 1.0, respectively. (**C**) Profiles of the individual quality scores used to filter the data. In order to ensure that the analyzed samples were not too diluted, the scores vary linearly between 0 (low quality) and 1 (high quality) for the intensity of the normalization peak ($Q_{norm}$) or the MBP-PDZ peak ($Q_{int}$). $Q_{2nd}$ is a quality score allowing to reject samples with low MBP-PDZ expression while $Q_{delta}$ combines the *TranslX* and *ScalX* parameters and varies exponentially.

For filtered data, the BI was extracted with the following equation (Eq 1) that estimates the depleted fraction after superimposition of the sample and reference electropherograms:

$$BI = \frac{I_{ref} - I_{lig}}{I_{ref}} \qquad (1)$$

where $I_{ref}$ and $I_{lig}$ are the intensities of the tag-PDZ peaks measured, for each PDZ domain/PBM peptide interaction pair, in the biotin reference and the PBM electropherograms, respectively.

Data reproducibility has been previously explored for several PDZ/PBM pairs resulting in a standard error of the mean of about 0.07 BI unit [31]. In some cases, reproducible negative BI values can be observed, for instance for MAGI2-1 binding to RSK1 PBM (BI = -0.09 ± 0.06) and for MPP4-1 or APBA3-2 binding to RSK1 phospho-PBM (BI = -0.16 ± 0.02, -0.14 ± 0.02, respectively), [16] as well as for PDZD7-1 binding to HPV16_E6 PBM (BI = -0.18 ± 0.02, a particularly low value) [31]. This likely result from a lower $I_{ref}$ intensity as compared to $I_{lig}$, potentially due a preference of the PDZ domain for beads fully saturated with biotin as compared to beads with biotinylated peptide. As reported previously, we have also investigated the

limit of detection by repeating the holdup experiments for an irrelevant "neutral" peptide owing no specific PBM consensus sequence. Almost all BI values were below 0.10 (98% of all measured PDZ/PBM pairs) and showed a standard deviation of less than 0.10 (considering 95% of the data) [31]. According to this, we considered a cut-off for BI of 0.20, representing a very stringent threshold that retains only high-confidence PDZ/peptide interactions, and eliminates most of the false positives. Altogether, these information are useful to grasp the quality of the holdup data.

## Steady-state fluorescence polarization

FP data were measured in 384-well plates (Greiner, Frickenhausen, Germany) using a PHER-AstarPlus multi-mode reader (BMG labtech, Offenburg, Germany) with 485 ± 20 nm and 528 ± 20 nm band-pass filters for excitation and emission, respectively. N-terminal fluorescein-labeled PBM peptides (f16E6, fRSK1 and fpRSK1) were used as tracers in direct and competitive FP assays. In competitive measurements, the 50 nM fluorescent reporter peptide was first mixed in 20 mM HEPES pH 7.5 buffer (containing 150 mM NaCl, 0.5 mM TCEP, 0,01% Tween 20) with the PDZ domain at a sufficient concentration to achieve high degree of complex formation (>60–80%). Subsequently, competitive FP assays were carried out by adding increasing amount of unlabeled peptide to the pre-formed complex with a total of 8 different peptide concentrations (including the 0 nM peptide concentration i.e. the absence of unlabeled peptide). Direct and competitive titration experiments were carried out in triplicate. The average competitive FP signal was used for fitting the data to a competitive binding equation with ProFit, an in-house Python-based program [41], allowing to extract the apparent affinity values. In our FP assays, every tested PDZ domain detectably bound to at least one PBM peptide, guarantying that PDZ domains are functional.

## Conversion from BI values to dissociation equilibrium constants

BIs were transformed into dissociation constants ($K_D$) using the following formula:

$$K_D = \frac{([PDZ_{tot}] - BI \cdot [PDZ_{tot}]) \cdot ([PBM_{tot}] - BI \cdot [PDZ_{tot}])}{BI \cdot [PDZ_{tot}]} \qquad (2)$$

where $[PDZ_{tot}]$ and $[PBM_{tot}]$ correspond to the total concentrations of the PDZ domain (usually around 4 μM) and the PBM peptide used during the assay. Since the $PBM_{tot}$ concentration in the resin during the holdup assay parameter may differ from one peptide to another and remains unknown, it is impossible to directly convert BI values into $K_D$ constants. We systematically determined the $K_D$ constants for a subset of PDZ/PBM pairs by competitive FP. These affinities were subsequently used to back-calculate the PBM peptide concentrations in the holdup assays according to Eq 2 when quantifiable and significant (>0.20) BI values were available for the same pairs. For each PBM, the average peptide concentration was calculated after outlier rejection based on the absolute distances from the median as compared to three times the standard deviation (3σ rule), with never more than 2 values rejected.

# Results

## An experimental strategy to measure large numbers of reliable affinity data

For this study, we wished to generate accurate and complete PDZome-binding affinity profiles for four peptide variants of the C-terminal PBM of PTEN. In practice, this requires measuring the individual affinities of 4x266 = 1064 distinct PBM-PDZ pairs. Taking into account the additional ~360 biotin-PDZ negative control measurements required for data treatment, the

assay represents ~1400 filtrates of protein extracts, which must each be individually subjected to capillary electrophoresis. Next, individual electropherograms must be visually curated and analyzed by an expert user to extract the binding intensities (BI) values that will compose the final profiles. Since the assay requires expensive materials and labor-intensive data treatment, we favored an approach based on singlicate holdup runs. Representative holdup data recorded for three PDZ domains (MAST2-1, HTRA1-1, SCRIB-3) with one PBM (PTEN_11) are shown in Fig 3A. After normalization of the two capillary electropherograms recorded for both the PBM of interest and the biotin reference, the comparison of the intensities of the two resulting PDZ peaks informs about the strength of the interaction: the stronger the depletion, the stronger the binding.

As an orthogonal unbiased high-throughput approach, competitive Fluorescence Polarization (FP) method has also been used to monitor the interactions of the same PDZ domains with PTEN_11 (Fig 3B). For the tested PDZ/PBM pairs, the apparent affinities were obtained by fitting the polarization data considering a competitive binding model [42]. The holdup BI values and the binding strength derived from competitive FP measurements are consistent: higher the BI, stronger the affinity. This trend suggests the possibility for PTEN data to cross-validate holdup data by competitive FP assay, as previously described for the PBMs of HPV16 E6 viral oncoprotein and RSK1 kinase [4].

## Generating PDZome-binding BI profiles of the four PTEN variant PBMs by holdup assay

We applied the holdup assay to generate PDZome-binding profiles of three 11-mer peptides (PTEN_11 for the native sequence, PTEN_Ac and PTEN_KR for the acetylated and K402R mutated version of PTEN_11, respectively), as well as an extended 13-mer peptide (PTEN_13). As described in the material and methods section, we rationalized the data curation step by introducing a numerical global quality score. We managed to quantify the interactions of 213, 233, 216 and 258 PDZ for the PTEN_11, PTEN_Ac, PTEN_KR and PTEN_13 peptides, respectively, which corresponds to 80%, 81%, 88% and 97% of the human PDZome. All holdup plots for which a binding intensity BI>0.20 has been detected, are shown in S1 Fig. The four resulting holdup datasets were then plotted independently in the form of "PDZome-binding profiles" representing the individual BI values versus the PDZ domains ranked from higher to lower BI values (Fig 4). PTEN_11 showed a maximal BI = 0.71, i.e. a significantly lower value as compared to the ones of PTEN_KR, PTEN_Ac or PTEN_13 (BI = 0.86, 0.90 and 0.81, respectively) when considering a BI standard error of ~0.07 BI unit [31]. Using BI>0.20 as a minimal threshold value for retaining high-confidence PDZ/peptide interactions, the holdup assay identified 19, 43, 37 and 24 PDZ domains as potential binders for the PTEN_11, PTEN_Ac, PTEN_KR and PTEN_13 peptides, respectively. Altogether, they represent a total of 123 potential binders, of which 60 are non-redundant PDZ domains distributed over 46 distinct proteins.

## Orthogonal validation by competitive FP and conversion of holdup BI data into dissociation constants of the four PTEN PBMs versus the human PDZome

Recorded in singlicate, the holdup data were systematically cross-validated for a subset of 20 PDZ domains by competitive FP assay (S2 Fig). The PDZ domains were selected so that each half of this subset was representative of either PBM binders, or non-binders, as detected in holdup data. Competitive FP assay has been chosen because it requires only several peptides to

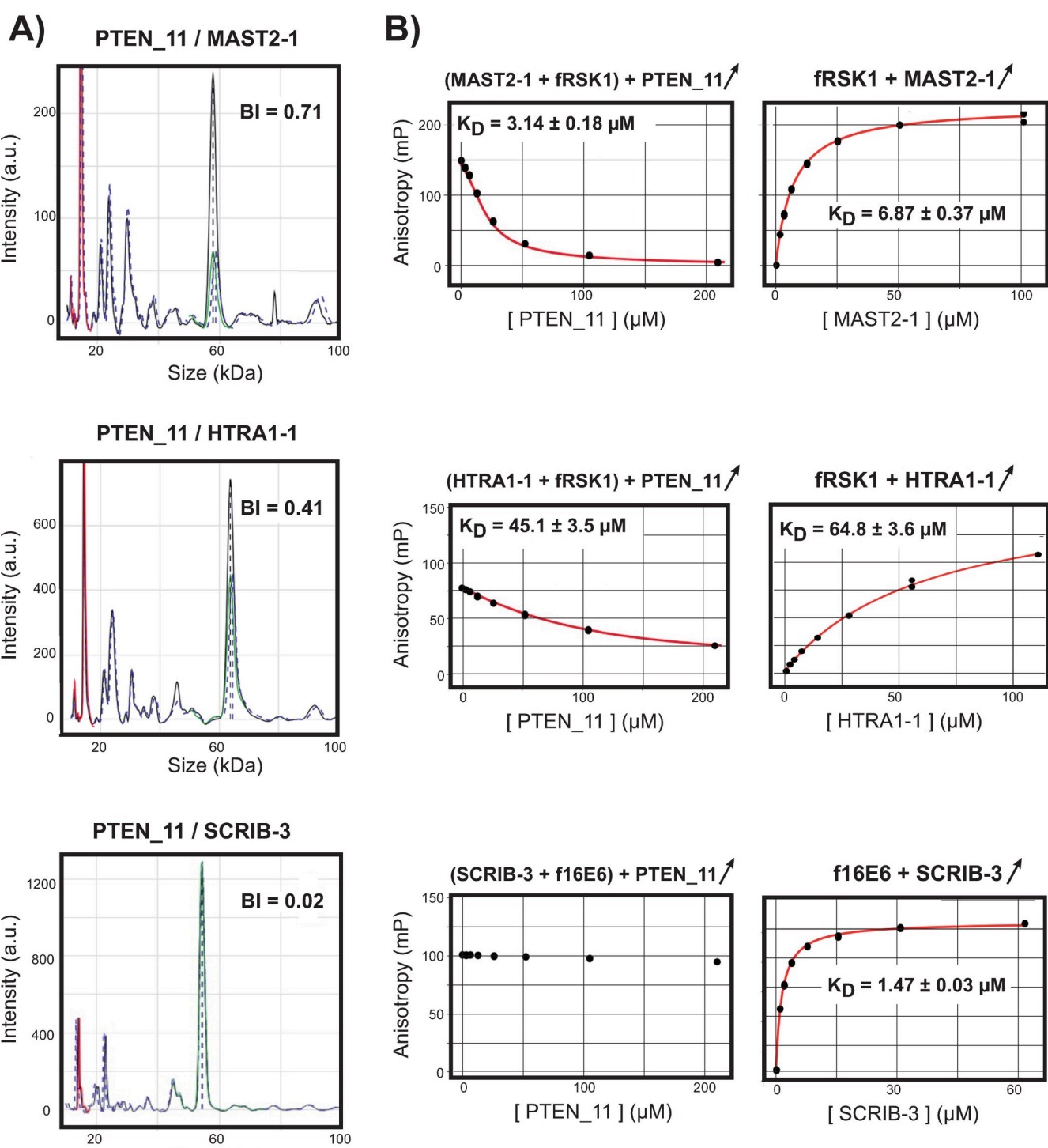

**Fig 3. Complementarity of holdup and fluorescent polarization data.** The interaction data of PTEN_11 with MAST2-1, HTRA1-1 and SCRIB-3 are shown as examples of strong affinity, weak affinity or non-binding, respectively, all measured by holdup (**A**) and competitive FP (**B**) methods. (**A**) After superimposition of the electropherograms recorded for the PBM of interest (blue dotted line) and for the biotin reference (black solid line), the normalization of the electropherogram of the PBM compared to the one of the reference is done using the signal of the lysozyme added in every sample at a constant concentration (red peak). The region between 20 and 60 kDa, which contains peaks of the crude extract supposedly to be constant, is used to verify the proper intensity normalization of the two electropherograms. The two intensities of the peak of interest read in the PBM and reference electropherograms after proper alignment along the molecular weight scale (region covered by the green dotted line) are subsequently used to quantify the depletion of an individual PDZ domain. All those normalization and alignment steps are performed automatically and are critical as the overlap of the two electropherograms is never perfect.

The holdup ultimately delivers "binding intensities" (BI) for each PBM/PDZ interaction pair, which in principle vary in a range from 0.00 (no binding) to 1.00 (strong binding). (**B**) Competitive FP measurements need a solution of pre-formed PDZ/labeled peptide complex titrated with increasing amounts of unlabeled peptide. Preliminary direct FP measurements recorded for the labeled peptides with increasing amounts of PDZ domains are visible on the right-hand column, showing the good quality of the chosen tracers. The pre-formed complexes used in competitive FP assays (middle column) consisted of MAST2-1 or HTRA1-1 mixed with 50 nM of labeled fRSK1 peptide, while SCRIB-3 was mixed with 50 nM f16E6 peptide. The PDZ concentration depends on each sample and was adjusted to reach >60–80% complex formation to ensure a satisfactory signal-to-noise ratio. Each panel shows the average of three competitive titration curves (black dots) and the fit results (red curves with the apparent $K_D$ values) using direct (right column) or competitive (middle column) binding model.

be labeled, and minimizes the detection of false-positive partners, as compared to direct FP measurements [41]. The PBM peptide tracers to be labeled (f16E6, fRSK1 and fpRSK1) were picked from our PDZ/PBM data base so that each of the 20 PDZ present in the subset was targeted by at least one of those three peptides. In addition each PDZ interaction with the PBM tracer was first confirmed by direct FP measurements. A scatter plot of experimental $K_D$ obtained by competitive FP ($K_D$_FP) *versus* BI shows a strong agreement between the holdup BI values and the binding strength derived from competitive FP measurements (S3 Fig). It confirms that a strategy combining holdup assay run in singlicate with competitive FP protocol run on a large proportion of the PDZ/PBM interacting pairs detected in the holdup assay, warrants the acquisition of highly reliable affinity data for all PDZ/PBM pairs that pass the quality score filtering step.

We were then interested to compare the data sets obtained for the different PTEN PBMs. However, this comparison implies the data to be expressed using an universal scale, as the binding free energy ΔG. Calculation of an equilibrium constant for a PDZ-PBM interaction requires three concentrations: free PBM, free PDZ and PDZ-PBM complex. The holdup assay delivers for each PDZ/PBM pair the concentrations of free PDZ and PDZ-PBM complex (obtained in the biotin and PBM electropherograms, respectively), but not that of free PBM., On the other side, the competitive FP resulted in approx. 8 to 10 significant $K_D$ for each PBM. These accurate dissociation constants were therefore used to back-calculate the peptide concentrations in the holdup assays (Fig 5A). We found the PBM peptide concentrations for the individual competitive runs to vary between 10 and 90 µM, with averages after outlier rejection between 17 and 34 µM depending on the PBM. A global mean of 26 µM considering all the peptides was determined. The (BI, $K_D$_FP) scatter plot described previously superimposed well with the theoretical affinity values calculated using the two extreme average peptide concentrations (S3 Fig), confirming that the back-calculated peptide concentrations are consistent with experimental data.

Using the mean concentration obtained above for each of the different PTEN peptides, the experimental BI values recorded by holdup for all tested domain/peptide pairs were subsequently transformed into equilibrium dissociation constants. A strong agreement is observed between the two sets of affinity constants obtained from holdup and competitive FP assays with a coefficient of determination $R^2$ = 0.74 (Fig 5B), confirming that singlicate holdup runs provided highly reliable data. Thereafter, affinity data measured by FP assay were also included in the FP data set for the PDZ domains (MAST1-1, MAST2-1, SNX27-1, MAGI1-2 and GRID2IP-2) for which holdup data were missing according to the low quality score in the filtering step. These interaction data detected only by competitive FP and not by holdup represented 1 to 2 additional PDZ binders per PTEN construct. A total of 215, 234, 218 and 259 interaction data were obtained for PTEN_11, PTEN_Ac, PTEN_KR and PTEN_13, respectively. This transformation into affinity values makes then possible to compare binding affinity profiles obtained for different peptides and potentially different batches.

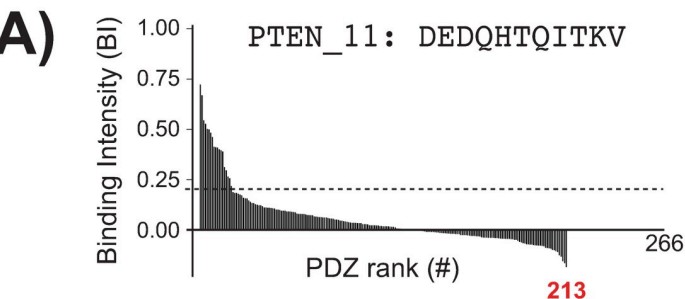

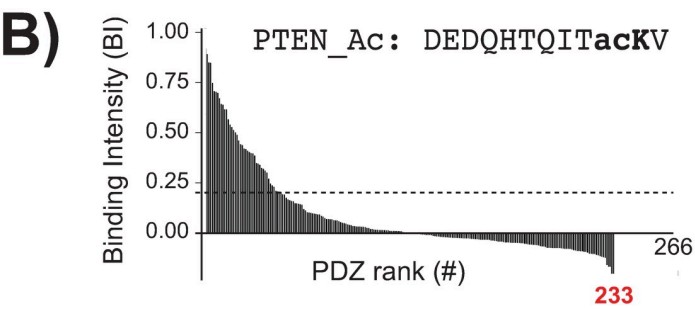

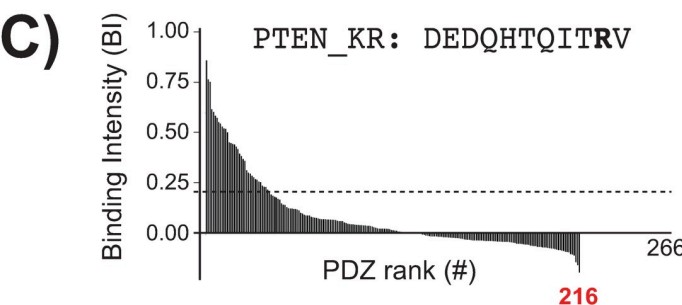

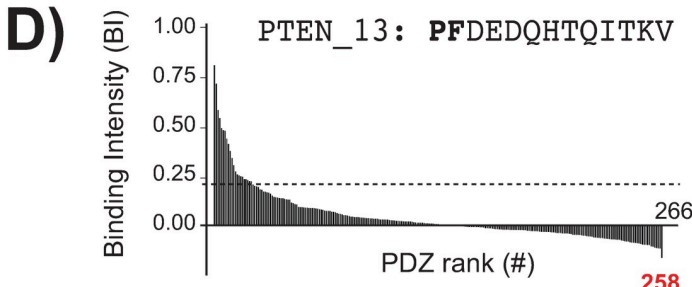

**Fig 4. PDZ binding profiles of the four PTEN peptides.** Holdup binding profiles obtained are shown for PTEN_11 (**A**), PTEN_Ac (**B**), PTEN_KR (**C**) and PTEN_13 (**D**). In each profile, the PDZ binders are ranked from left to right of the plot in BI decreasing order along the X-axis. All the measured holdup data are shown. The grey dotted line shows the threshold for confidence value, set at BI = 0.20 (see main text). For each experiment, the number of PDZ domains

for which we obtained a measurement that passed the quality filtering step, and could therefore be included in the plot, is indicated (red case numbers). All the holdup data for PDZ/PBM pairs with BI>0.20 after processing are shown in S1 Fig.

### From binding profiles to specificity quantification

The above described holdup-FP strategy delivers binding affinity constants, a universal chemical property. The affinity values obtained for each PTEN peptide were plotted in a logarithmic scale, hence proportional to free energies of binding ΔG at a fixed temperature (Fig 6). The resulting profiles contains information about specificity, or promiscuity, since a promiscuous peptide as seen by holdup would bind to a large number of PDZ. We then looked for a numerical parameter that would express, in a quantitative way, this specificity or promiscuity information. For this purpose, we calculated for each profile the difference between the maximal and the minimal significant affinity values detected by the assay, $\Delta G_{max} - \Delta G_{min}$. Next, we introduced a threshold affinity, called "half-maximal binding affinity" defined as follows:

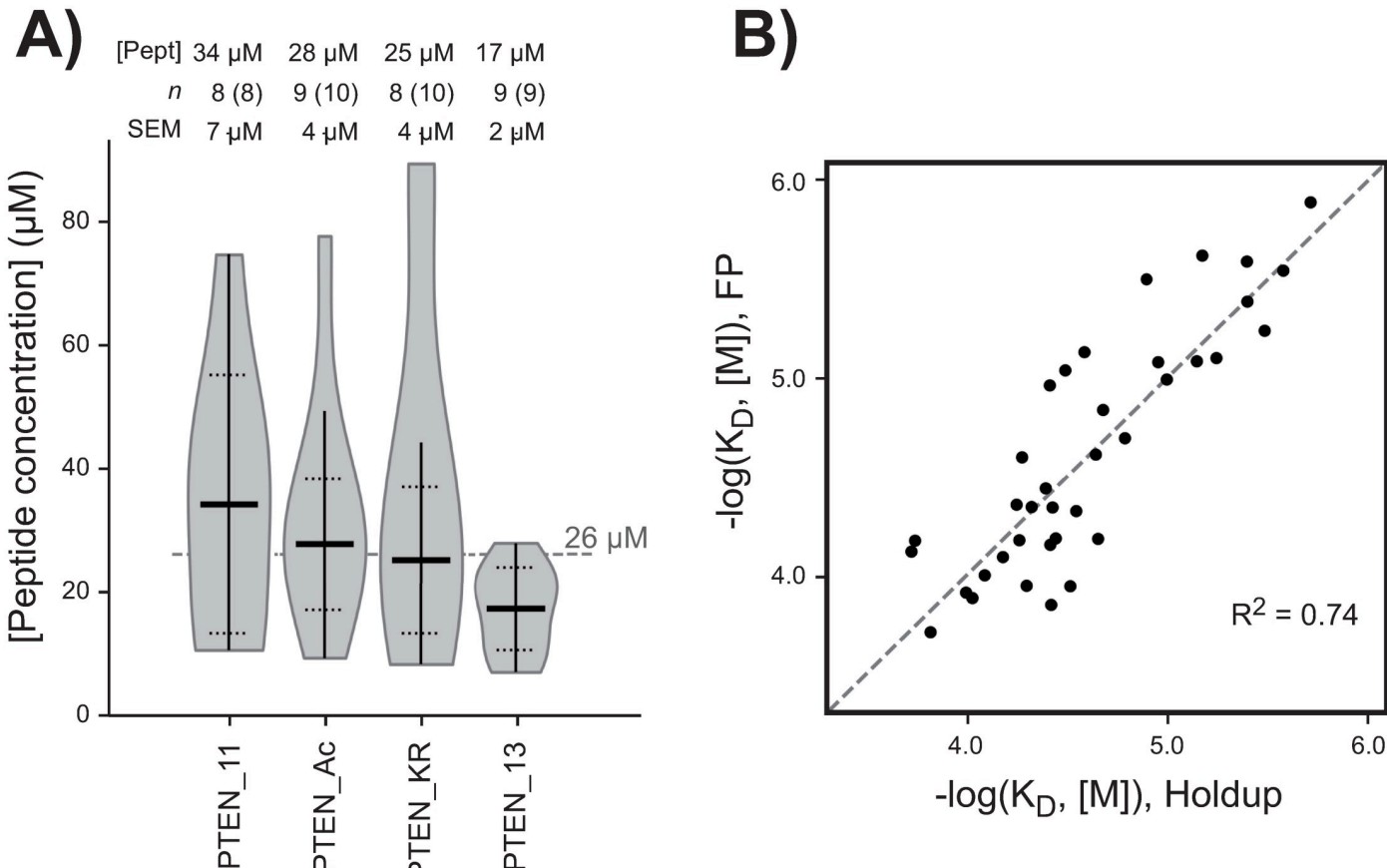

**Fig 5. Conversion of the holdup binding intensities into affinities constants.** (**A**) The violin plot shows the distribution of all the back-calculated apparent peptide concentrations obtained when both a quantifiable and significant (>0.20) BI value by holdup and a dissociation constant by FP were available for a given PDZ/PBM pair. On each violin representation, the vertical line indicates the range of the distribution while the horizontal lines show the final mean peptide concentration and its final standard deviation after outlier exclusion (considering the 3σ rule). The final average peptide concentrations represented by the thick lines are used to convert the holdup BI values into $K_D$. On the top are given the peptide concentrations, the numbers of data points with and without exclusion and the standard errors of the mean. (B) Comparison between the converted dissociation constants from the holdup assay and the dissociation constants directly measured by FP assay. The dotted line represents the perfect theoretical correlation. Since the data points are randomly distributed on both sides of this dotted line, the $R^2$ is indicative of the goodness of fit.

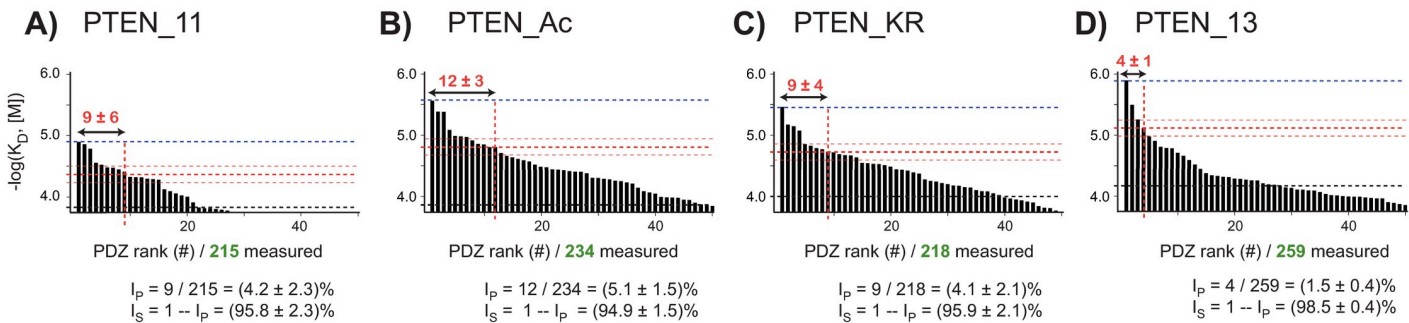

**Fig 6. Determination of the specificity index for the PTEN binding profiles.** For every profile, the significant PDZ binder affinity values are ranked from left to right along the X-axis in -log($K_D$) decreasing order. The non-significant or undetected binders were omitted for clarity. The grey dotted line corresponds to the threshold BI value after converting it into -log($K_D$) scale, while the blue and red thick dotted lines represent the highest affinity and the affinity at half the difference between the maximal and weakest significant affinity values, respectively. The reader can note that, for a constant threshold BI value (0.20), the weakest affinity values may vary moderately due to non-constant peptide concentrations. The numbers of PDZ domains above the half-maximal binding affinity" are indicated in red, while the numbers of tested and validated PDZ domains are in green. Values calculated for the promiscuity index ($I_P$) and the specificity index ($I_S$) are given (see main text). The overall uncertainty on log($K_D$) values was estimated to be roughly ± 0.12 in log(M) unit and is illustrated by thin red dotted lines. Full data sets for holdup and FP are visible in S1 and S2 Figs, respectively.

$\Delta G_{half} = \Delta G_{min} + (\Delta G_{max} - \Delta G_{min})/2$. We then defined the promiscuity index $I_P$ as the percentage of PDZ domains bound to the PBM with an affinity superior to the half-maximal affinity, relative to the total number of PDZ domains that were successfully measured in the assay (Fig 6). Alternatively, the specificity index $I_S$ could be defined as $1 - I_P$. Therefore, the lower the promiscuity index, the higher the specificity index, the higher the specificity of the PBM for a limited number of selected domains across the PDZome. For instance, if 250 PDZ domains were fully assayed, and only 5 PDZ domains bound to the PBM with an affinity superior to the half-maximal affinity, the specificity index will be 98%. If 25 domains bound with an affinity superior to the half-maximal affinity, the specificity index will be 90%.

We probed the specificity index on the PDZome-binding profiles of the four PTEN peptides. In both the BI-based and the affinity-based representations (Figs 4 and 6), the shapes of the profiles of PTEN_11, PTEN_Ac, PTEN_KR look similar, while the PTEN_13 seems to present a sharper, faster decreasing profile. This would indicate, in qualitative terms, that the PTEN_13 peptide selects PDZ domains in a more specific -less promiscuous- way than that of the three of other peptides. This observation is in line with the computed specificity indexes: although the index values are highly similar between PTEN_11, PTEN_Ac, and PTEN_KR [(95.8 ± 2.3)%, (94.9 ± 1.5)%, (95.9 ± 2.1)%, respectively], the extended wild-type peptide PTEN_13 displays a slightly higher specificity index [(98.5 ± 0.4)%; p-value = 0.039 for PTEN_Ac vs. PTEN_13 considering a Fisher's exact test] which could be indicative of a higher specificity towards several selected PDZ domains. One should note that it was not possible to determine index uncertainty by error propagation starting from holdup data, since BI values have been recorded in singlicate. However, the dispersion of log($K_D$) values around the perfect theoretical correlation line in the correlation plot (Fig 5B) is representative of global $I_S$ uncertainty, which was therefore estimated by averaging the distances between the experimental dots and the theoretical diagonal. Details about uncertainties and Fisher's test calculations are provided in Supp. Materials and S1 File.

### Rearrangements of the binding profiles due to minor changes in PTEN

The PTEN-bound PDZ domains are distributed over a diversity of PDZ-containing proteins (Fig 7). Several PDZ domains such as MAST2-1, PDZD7-3, SNX27-1, MAGI1-3 and GRASP-1 were systematically found among the strongest interaction partners of all four PTEN PBM

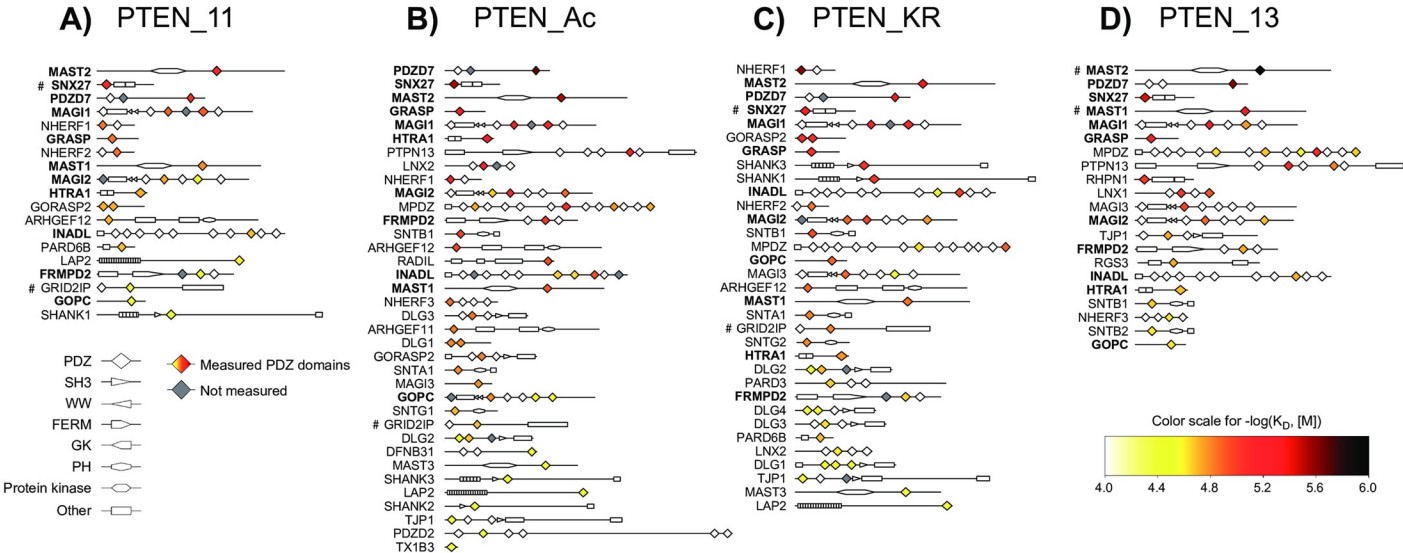

**Fig 7. Affinity-based heat maps of the impacted PDZ domains by the different PTEN peptides in the context of full-length proteins.** Proteins containing PDZ domains significantly bound to one PTEN peptide are colored and ranked from strongest to weakest binding strength depending on the best individual PDZ binder within each protein. The color code from white to black is indicative of the -log($K_D$) values in the range of 4.00–6.00 after filtering step and BI conversion. The symbol (#) denotes PDZ domain for which the BI value could not be measured directly by holdup and has been inferred from FP measurements. Protein names appeared in bold when significant–log($K_D$) values are observed for the four PTEN PBM. Numerical BI and -log($K_D$) values can be found in the S1 File.

variants. We compared our data to previously published studies (Table 1). Bearing in mind that sequences and boundaries of PTEN and PDZ constructs may differ, our results agree with isothermal titration calorimetry data obtained for SNX27-1/PTEN [43] and MAST2-1/PTEN complexes [29] and, in part, with competitive FP data obtained for PARD3-1/PTEN complex [44]. Interestingly, some of our newly identified PTEN-binding PDZ domains, such as MAGI1-3, MAGI2-3 and DLG4-1 bound wild-type PTEN peptides with a stronger affinity than the domains of the same proteins that were previously published to bind PTEN, such as MAGI1-2 [45], MAGI2-2 [26] and DLG4-3 [46], respectively. This observation illustrates the strength of the complementary holdup/FP approach which can provide an affinity ranking of PDZ domains even within multi PDZ-containing proteins.

Although the shapes of the dissociation constant profiles for the three 11-mer PTEN variants were globally similar, the PDZ domains are reshuffled between the various profiles (Fig 8). We detected at least 20 additional new partners for PTEN_Ac, and 11 for PTEN_KR (Fig 8A and S1 File). The acetylated peptide is highly promiscuous and binds to all the partners of the native PTEN_11 PBM, plus numerous additional ones. Furthermore, the arginine mutation does not seem able to efficiently reproduce the acetylated state as seen by the number of partners (8 over a total of 37) detected for PTEN_KR and not for PTEN_Ac. The opposite effect with 15 over a total of 43 detected for PTEN_Ac and not for PTEN_KR is even more pronounced, suggesting that the acetylation effect on binding is mainly due to the acetyl group rather than the size of the side chain carried by the acetylated lysine residue.

The fourth PDZ domain of PTPN13, a protein tyrosine phosphatase enzyme, appears among the 20 new partners detected only for PTEN_Ac. Although a binding between PTEN and the first and/or second PDZ of PTPN13 has been described previously, the same authors observed the absence of interaction mediated by PTPN13-4 domain with PTEN [28]. Our data shows, not only that PTPN13-2 is interacting with the wild-type PTEN PBM and not with the acetylated version of the PBM, but also that PTPN13-4 binds to PTEN_Ac. It has been shown

**Table 1. PDZ domains interactors for PTEN according to literature and the present study.**

| Protein [a] | PDZ dom [b] | Method [c] | Ref | $K_D$ [d] | PTEN_11 [e] | PTEN_Ac [e] | PTEN_KR [e] | PTEN_13 [e] |
|---|---|---|---|---|---|---|---|---|
| DLG1 | 2 | Pull-down | [25] | | nd | 36 | 71 | 81 |
| DLG4 | 1 | | | | 293 | 155 | 81 | nd |
| | 3 | Co-IP | [46] | | nd | nd | nd | nd |
| MAGI1 | 2 | Co-IP | [45] | | nd | nd | nd | nd |
| | 3 | | | | 30 | 10 | 15 | 10 |
| MAGI2 | 2 | Pull-down, IP | [26] | | 152 | 56 | 29 | 149 |
| | 3 | | | | 47 | 15 | 21 | 30 |
| MAGI3 | 2 | Pull-down, Co-IP | [27] | | nd | 39 | 29 | 21 |
| MAST1 | 1 | Pull-down, Co-IP | [47] | | 39 | 26 | 32 | 8 (*) |
| MAST2 | 1 | ITC | [29] | 2 | 13 | 4 | 7 | 1 (*) |
| MAST3 | 1 | Pull-down | [25] | | 241 | 85 | 82 | nd |
| NHERF1 | 1 | Pull-down, Co-IP | [48] | | 33 | 14 | 4 | nm |
| NHERF2 | 1 | Co-IP, Pull-down, Overlay assay | [24] | | nd | nd | 120 | 134 |
| | 2 | | | | 35 | nd | 21 | 125 |
| PARD3 | 3 | FP | [44] | 19 | 160 | nd | 56 | 96 |
| PTPN13 | 2 | Pull-down | [28] | | 148 | nd | 153 | 16 |
| | 4 | | | | nd | 10 | nd | 36 |
| SDCBP | 1 | LC-MS | [49] | | nd | nd | nd | nd |
| SNTB2 | 1 | LC-MS | [49] | | nd | nm | nd | 64 |
| SNX27 | 1 | ITC | [43] | 38 | 14 (*) | 4 | 8 (*) | 6 |

Each row corresponds to a protein for which a binding to PTEN has been described in literature. The main methods and the PDZ domain number are indicated. The four last columns contain information obtained by combining the holdup and FP data presented in this study.

[a] Protein name.

[b] Domain interaction site for PTEN.

[c] Detection methods described in literature.

[d] Affinity provided in the literature when available (in μM).

[e] Affinity measured by holdup in this study (in μM).

[*] Affinity measured by FP in this study (in μM).

IP: Immunoprecipitation.

Co-IP: Co-immunoprecipitation.

nd: Not detected in the holdup assay.

nm: Not measured in the holdup assay.

that the binding of the tumor suppressor PTEN, through its PBM, to PDZ domain-containing partners is a major mechanism of PTEN subcellular targeting and protein stabilization [23]. Furthermore, a tumor suppressive role has been suggested for the PTEN PBM in a breast cancer model [50]. Therefore, the reorganization of the cellular targets of PTEN PBM as we observed may have a deep impact on its biological activities with potential relevance in tumor suppression and cell homeostasis.

The impact of the PTEN peptide length was noticeable by comparing the dissociation constant profiles of PTEN_11 and PTEN_13 (Fig 8B). The detected interactions of PTEN_13 were markedly stronger compared to the affinities observed for the same PDZ partners in PTEN_11. The strongest effect is observed for MAST2, the top binder for both PTEN_11 and PTEN_13, for which the–$\log(K_D)$ value increases from 4.90 ± 0.12 to 5.89 ± 0.12 in log(M) unit (i.e. a jump from 13 μM to 1 μM), corresponding to about a 10 fold stronger affinity. Only a limited number of interactions, in the low range affinities, were potentially slightly

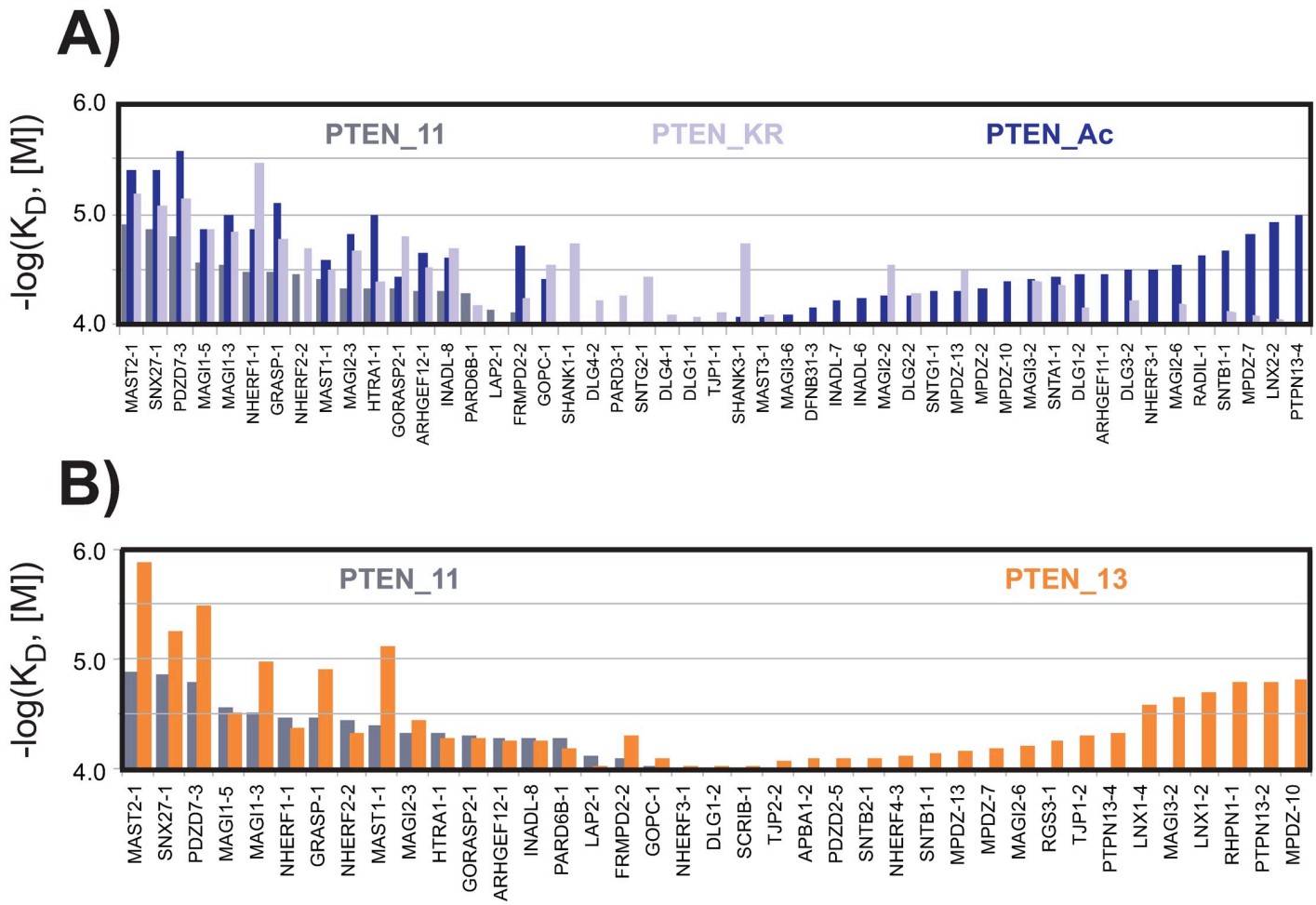

**Fig 8. Changes in the PDZ binding profiles induced by changes in the PTEN peptides.** (**A**) Comparison between PTEN_11 (grey), PTEN_KR (light purple) and PTEN_Ac (dark blue) using a shared PDZ axis. For the wild-type PTEN_11 peptide, the PDZ domains were ranked in descending affinity order along the X-axis, from left to right according to the significant affinities for PTEN_11, and from right to left according to the significant affinities solely detected for PTEN_13. The remaining PDZ domains that bind only to the PTEN_KR peptide were added in the middle region. (**B**) Comparison between PTEN_11 (grey) and PTEN_13 (orange) on a shared PDZ axis. The PDZ domains were ranked along the X-axis in descending order, from left to right according to the significant affinities for PTEN_11, and from right to left according to the significant affinities exclusively detected for PTEN_13. The left and right regions thus show PDZ domains that prefer the shorter or the longer PTEN PBM version, respectively. The overall uncertainty on $\log(K_D)$ values was estimated to be roughly ±0.12 in $\log(M)$ unit.

strengthened, although most likely not significantly. Moreover, 24 new binders appear apparently due to the presence of the two extra residues in the N-terminus of the peptide. Altogether, the rearrangements observed in the present work are particularly noteworthy since the mutations or the Pro-Phe inclusion are located at positions described as non-critical for PBM binding.

## Discussion

### Insight into the holdup: A powerful semi-automated tool for medium-to-low affinity measurements

In this work, we quantitatively assessed more than 1,000 distinct PDZ-peptide affinities by using a "crude holdup assay" protocol, which quantifies the disappearance of a single protein peak (the tag-PDZ peak) out of a complex crude overexpression extract. This protocol requires

a rigorous approach. Some critical biochemical steps have been previously identified [31,32] including the standardized expression of the complete PDZome, the verification of its quality, the calibration of its concentrations in the crude extract, and a careful quality control of capillary electrophoresis runs. For data treatment, we developed a computational processing step for accurate superimposition of the electropherograms to improve the precision of binding intensities [38]. Here, a four-criteria quality score was introduced to further rationalize data curation. These improvements allow us to minimize the amount of false positive and false negative results. In addition, to spare costs and manpower for data treatment, holdup experiments were run in singlicate and combined with an orthogonal approach, the competitive FP. This generated high-confidence affinity data and allowed us to convert holdup binding intensities (BI) values into affinities ($\Delta G$ or $K_D$). The use of such an intrinsic universal parameter of molecular complexes also presented the advantage to facilitate the comparison with data available in the literature. In future developments of the automated holdup assay, we envision to replace crude overexpression extracts by purified proteins, which greatly facilitate both read-out and data treatment [34].

## Impact of PTEN PBM acetylation on its PDZ interactome

Lysine acetylation is a PTM difficult to study and reproduce *in vitro*. Some studies have explored lysine acetylation by proteomic approaches [51], while others have mutated lysine residues to glutamine or arginine to mimic acetylation or suppress acetylatability, respectively [22,52–54]. In the present study, we investigated with chemically synthetized peptides that allow to fully control PTM the differential effects of acetylation or mutation of a lysine residue on the PDZ interactome of PTEN. PTEN is a tumor suppressor that is frequently inactivated in human cancers [55,56]. Some *in vivo* activities of PTEN such as PI3K signaling regulation seem to be abolished when PTEN is acetylated [57]. In addition, the Lys-to-Arg mutation at PTEN position 402 (corresponding to a non-essential p-1 position of its C-terminal PBM) abolished PTEN acetylatability [22]. However, this may either mean that K402 is a direct acetylation target or indicates that the integrity of the PTEN PBM sequence is required for PBM-dependent acetylation of PTEN at other sites distinct from K402. We found that K402 acetylation (inducing a loss of a positive charge and a slight increase of bulkiness) altered both the strength and the number of detected PDZ binders of PTEN. In contrast, the K402R mutation (preserving the positive charge but further increasing the bulkiness) did not alter the overall binding strength nor the number of binders. Furthermore, the K402R mutant retains binding to most partners of the native motif and also binds to a subset of the acetylated peptide partners. Therefore, the presence or absence of a positive charge at the p-1 position of the PTEN PBM appears to be more critical for PDZ recognition than the bulkiness of the side chain. This is noteworthy since the p-1 position has been often described in literature as a non-critical position of the canonical PBM motif for PDZ/PBM interaction. One should note that our conclusion is in line with the work of Tonikian *et al.* [12]. Although our work focused on binding profiles for peptides originating from a given PBM (PTEN) while those authors analyzed by phage display the binding profiles for a given PDZ domain, the LAP2-1, they found in particular that mutations in the LAP2-1 PDZ domain specifically affect the binding preference within the PBM sequence, even at position p-1.

Although some PDZ domains including several ones from MAGI and NHERF detectably bound to all the three PTEN_11, PTEN_Ac and PTEN_KR peptides, several PDZ domains bound only one or two of those peptides. For instance, both PTEN_Ac and PTEN_KR bound stronger than wild-type PTEN_11 to MAGI2_2 or DLG1_2 domains, in agreement with Ikenoue *et al*. Since our study is covering the full PDZome, this implies that, for a majority of

PDZ domains, their binding affinity for PTEN was reinforced by acetylation. Overall, the rather large number of PDZ partners associating with the PTEN PBM confirms that domain/motif networks are rather promiscuous [58].

## Lessons from distal residues on the PTEN interactome

There is no consensus for the precise residue length of a given PBM needed to complete the interaction with a PDZ domain. Although the four C-terminal residues are usually thought to constitute the core of a PBM, it was shown that peptides encompassing the last 10 positions of a PBM undergo a significant change in their PDZ-binding affinities as compared to peptides comprising only the last 5 positions [14]. Such affinity variations may result from differences of entropy of the free peptides, from altered interface contacts in the resulting PDZ-PBM complexes, or a combination of both. Accordingly, synthetic or recombinant PBMs employed for PDZ interactions generally include at least 9 to 11 residues [4,5,12,16,18,31]. Indeed, the presence of distal sites altering PDZ-PBM binding has already been described [59], even at positions as far as at p-36 [60]. In the particular case of PTEN, Terrien *et al.* previously demonstrated the existence of a distal "exosite" at F392 (p-11), that triggers novel contacts within a secondary exposed hydrophobic surface of MAST2 [29]. Here, we showed that the inclusion of two extra residues, including F392, (PTEN_13 versus PTEN_11) affected both the PDZ interactome identified for PTEN and the specificity of its PBM. Indeed, several PDZ domains detectably bound only to the longer construct, in line with the idea of a global affinity increase because of the larger number of atomic contacts. Furthermore, while the three 11-mer peptides displayed equivalent PDZ-binding specificity, PTEN_13 showed a slight increased specificity. The addition of two extra residues corresponding to the distal exosite previously described, was therefore more influential for specificity than the chemical variations at p-1 position (Lys acetylation or Lys-to-Arg mutation).

In principle, one may argue that domain-motif binding events may be altered by any distal region, so that only studies of full-length protein/protein interactions are relevant. Notwithstanding the methodological issues (large full-length proteins can be very difficult to handle), one must keep in mind that most full-length multi-domain proteins are prone to many conformational changes (inducible by partner binding, ligand binding, PTM, molecular crowding, and so forth), which in turn influence the availability of their globular domains or linear motifs for binding events. This justifies the 'domainomics' approach [61] undertaken in this work, that focuses on the binding properties of minimal interacting fragments of proteins, such as a globular domains (e.g., PDZs) and short linear motifs (e.g., PBMs). Even if our binder list might be incomplete as compared to studies involving full-length proteins, it provides a list of the PDZ domains capable to interact with the motif of the PTEN PBM, constituting the minimal block at the binding interface of protein/protein interactions.

## To bind or not to bind

In this work, by covering almost the entire PDZ family, we quantified both the number of interacting and non-interacting partners for a given PBM. The knowledge of the two numbers is important since the count of 3 binding partners over a dataset of 10 domains, or 3 partners over a dataset of 100, is not reporting the same specificity. Over the years, we have accumulated holdup data for many peptides and noticed that more than 90% (244/266) of the PDZs in our expressed PDZome are functionally active since they interacted significantly with at least one PBM [32]. This indicates that most of the non-binders detected in validated holdup experiments and visible in our profiles are trustable. The holdup assay is therefore a reliable approach

to address not only the specificity but also possibly the 'negatome' in the sense of the negative interaction dataset as originally proposed [62].

Usually, specificity is evoked after comparing the affinity for a given target of one domain or protein with another or several others. In this work, we derived from the PDZome-binding profiles a unique numerical value, that we called the "specificity index", to evaluate the degree of specificity of a given PBM towards the PDZome. One may argue that the calculated specificity index is biased since our interaction datasets do not cover 100% of the PDZ domain family. However, we consider that the specificity indexes would be roughly the same for both the validated and the complete PDZ datasets if we assume the probability of binder occurrence to be even more similar in the validated and untested PDZ datasets as the validated dataset is covering a large part ($>\sim$80%) of the entire human PDZome. On the other side, one must notice that this index is not fully satisfying and cannot be considered as a universal parameter beyond our particular PBM-PDZome affinity profiling studies. In particular this index is only operative to compare profiles with a roughly continuous decreasing shape, e.g. in absence of discontinuous "breaks" or "stairs", which would introduce a large uncertainty of the index. But, to the best of our knowledge, the concept of specificity index constitutes the first attempt of a numerical value to describe binding specificity in a context of domain/peptide interactions covering a full domain family. It affords the advantage of introducing a numerical value attached to each PBM profile, that will ease their comparison. The knowledge of the specificity for different PBM, combined with structural and/or sequence analysis, may constitute an attractive possibility to define PTEN C-terminal mimicking peptides which could bind with high specificity to distinct PDZ domains and interfere with the formation of physiological PDZ/PTEN complexes.

## Conclusion

In this study, we showed that the addition of two extra residues representing a distal exosite and including a hydrophobic phenylalanine, not only impacts the interaction of the PTEN C-terminal tail with MAST2 as previously reported [29], but also affects its binding to a large set of other PDZ interaction partners, suggesting to well control the length of the polypeptide used for *in vitro* interaction studies. More importantly, we also showed that both, the K402 acetylation and the K402R point mutation at p-1, a non-critical position of the canonical PBM motif for PDZ/PBM interaction, significantly increased the number of targeted PDZ domains mainly by shifting the affinities toward stronger values although some exceptions have also been observed. This observation could be of primary relevance, knowing that the activities of the tumor suppressor PTEN protein is regulated by acetylation. Finally, we also introduced a way to quantify specificity that could be extended to other interaction studies covering a whole domain family.

## Supporting information

**S1 Fig. The entire data set obtained by holdup for BI$>$0.20.** For each panel, after superimposition of the two electropherograms recorded for the PBM of interest (blue dotted line) and for the biotin reference (black solid line), the normalization of the PBM electropherogram compared to the reference one is done using the signal of the lysozyme added in every sample at a constant concentration (red peak). The region between 20 and 60 kDa which contains peaks of the crude extract supposedly to be constant, is used to verify the proper intensity normalization of the two electropherograms. The intensities of the peak of interest after proper alignment along the molecular weight scale (region covered by the green dotted line) are subsequently used to quantify the depletion of an individual PDZ domain and then the BI

value. All those normalization and alignment steps are performed automatically.
(PDF)

**S2 Fig. The entire data set obtained by competitive FP.** The first column contains direct FP data, while the others contain competitive FP data. FP data recorded in triplicate are represented by black dots. The reported dissociation constants and errors are the averages and the standard deviations of the fit (solid red curves) of 500 independent Monte-Carlo simulations, calculated using ProFit as described in Simon et al., 2020.
(PDF)

**S3 Fig. Comparaison of holdup BI and $K_D$ values obtained by competitive FP.** The scatter plot corresponds to experimental holdup BI $K_D\_FP$ values colored according to the PBM peptides. Is superimposed the curves $K_D = f(BI)$ obtained from Eq 2 considering the global average peptide concentration (26 μM; black solid line) or the lower and upper peptide concentrations (17 and 34 μM, gray solid lines). Error bars are representative of peptide concentration uncertainties (See Fig 4) after their propagation into the $-\log(K_D)$ values.
(PDF)

**S1 File. All data set with all the BI values together with the transformed dissociation equilibrium constants for each PDZ-PBM interaction.** All the plots in the present study are obtained according to the data contained in this file.
(XLSX)

**S2 File.**
(DOCX)

## Acknowledgments

The authors thank Nicodème Paul for helpful discussions about statistical tests.

## Author Contributions

**Conceptualization:** Pau Jané, Gergő Gógl, Nicolas Wolff, Gilles Travé, Yves Nominé.

**Data curation:** Pau Jané, Gergő Gógl, Goran Bich, Yves Nominé.

**Formal analysis:** Nicolas Wolff, Gilles Travé, Yves Nominé.

**Funding acquisition:** Nicolas Wolff, Gilles Travé, Yves Nominé.

**Investigation:** Gergő Gógl, Camille Kostmann, Virginie Girault, Célia Caillet-Saguy, Renaud Vincentelli.

**Methodology:** Gergő Gógl, Pascal Eberling, Renaud Vincentelli, Yves Nominé.

**Project administration:** Gilles Travé.

**Resources:** Nicolas Wolff, Gilles Travé.

**Software:** Pau Jané, Yves Nominé.

**Supervision:** Nicolas Wolff, Gilles Travé, Yves Nominé.

**Validation:** Pau Jané, Gergő Gógl, Nicolas Wolff, Gilles Travé, Yves Nominé.

**Visualization:** Pau Jané, Gergő Gógl, Goran Bich, Yves Nominé.

**Writing – original draft:** Pau Jané, Gilles Travé, Yves Nominé.

**Writing – review & editing:** Gergő Gógl, Camille Kostmann, Goran Bich, Virginie Girault, Célia Caillet-Saguy, Pascal Eberling, Renaud Vincentelli, Nicolas Wolff.

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
