## [Decision Letter · Decision Letter 0]

4 Nov 2020

PONE-D-20-29437

Interactomic affinity profiling by holdup assay: acetylation and distal residues impact the PDZome-binding specificity of PTEN phosphatase

PLOS ONE

Dear Dr. Nominé,

Thank you for submitting your manuscript to PLOS ONE. After careful consideration, we feel that it has merit but does not fully meet PLOS ONE’s publication criteria as it currently stands. Therefore, we invite you to submit a revised version of the manuscript that addresses the points raised during the review process.

Please see the comments by the two referees, who both find your manuscript interesting and technically correct, but point out several minor points that should be addressed in a revised version. 

We look forward to receiving your revised manuscript.

Kind regards,

Petri Kursula

Academic Editor

PLOS ONE

Journal Requirements:

Reviewers' comments:

Reviewer's Responses to Questions

**Comments to the Author**

1. Is the manuscript technically sound, and do the data support the conclusions?

Reviewer #1: Yes

Reviewer #2: Yes

2. Has the statistical analysis been performed appropriately and rigorously? 

Reviewer #1: Yes

Reviewer #2: Yes

3. Have the authors made all data underlying the findings in their manuscript fully available?

Reviewer #1: Yes

Reviewer #2: Yes

4. Is the manuscript presented in an intelligible fashion and written in standard English?

Reviewer #1: No

Reviewer #2: Yes

5. Review Comments to the Author

Reviewer #1: The authors combined high-throughput holdup chromatographic assay and competitive fluorescence polarization technique to measure the binding affinity of the PDZ binding motif peptides with a PDZome including 266 PDZ domains. This strategy could be useful to measure other protein interactions. The authors developed a computational processing step to improve the precision of binding intensities. The authors concluded that the mutation at the distal region and acetylation of the core residue could alter the PDZ interaction. These findings make this article meaningful for publication in the journal. After carefully reading, some points which were found from this paper were listed below:

1) Line 36-37, “few” means little or none, “many” or “several” instead of "few"could be better here. The words “few” and “a few” were used many times in the paper, please choose proper word to describe the number.

2) The paragraphs (line 80-103) is to describe the PDZ binding motif, which is not easily understood by not experienced readers. If an extra figure concerning the alignment of the four PTEN peptides, highlight of the key residues, amino acid residues position, consensus sequence, were included in the paper, it would be much helpful.

3) Line 92, PTM should be listed in abbreviation.

4) The paragraphs (line 124-159) could be shorter. Some contents are duplicated with those in the method and result sections.

5) Please describe the plasmids, pETG41A and pETG20A(line 167)

6) Please describe MBP and TRX (line 168) and add them to the Abbreviation

7) Line 180: What do the bold letters mean? should be explained!

8) The source of the peptide sequences should be described. The figure mentioned in the point 2 could be included in this section (line 179-190)

9) Please explain the three PDZ domains in the main text (line 323)

10) This paragraph (line 358-360) should be mentioned in the section of the method

11) Line 438, this sentence was not precisely described.

12) Line 548 -549, this sentence is not complete, could cause misleading.

13) Line 568-571, is the two extra residues at the C-terminus of PTEN_13 from the wild type sequence of PTEN or mutation? It should be described in the text. If mutation, 13mer PBM instead of PTEN_11 should be used as a reference.

14) Line 613: This sentence is not precise.

15) Line 621: The acetylation not only increase, but also reduce the affinity of PTEN for PDZ domains (See Fig. 7, Table 1).

16) Line 699, not always increased, sometimes lost binding affinity (See Fig. 7, table 1)

17) In Fig.6., the color scale is not so clear, difficult to be compared. If one column with exact number for -log(KD,(M), it would be much better.

Reviewer #2: This work from Jané et al. deals with a high-throughput assay aiming the specificity profiling of the whole set of the human PDZ domains versus four variants of the C-terminal region of PTEN protein. Results confirmed that the acetylation of a lysine residue (a PTM that seems to occur in vivo with functional relevance) significantly alters the binding profile. The manuscript is well written, and it encompasses a nicely performed and vast amount of work. I think that it can be published in Plos One with just minor points to be revised. Concretely:

1. There is a literature reference that should be addressed (Tonikian, R. et al. A Specificity Map for the PDZ Domain Family. PLoS Biology 2008, 6, e239), where a specificity map of PDZ domains is offered. The reference should be included both in the introduction (lines 88-91) and especially in lines 612-616, which are located in the discussion. In Toninkian’s paper, position -1 is also relevant in specificity binding to PDZ domains, and it should be at least commented.

2. Having checked reference #31, I am not sure whether the cell lysates are purified based on a His-tagged chromatography or another method or if the lysates are directly used. In the latter case, it seems to me strange to use direct cell lysates. I would like this to be clarified.

3. In lines 483-489, there should be a nuance in the comparison of the specificity indexes of the four peptide variants. The percentages are indeed very similar between them (96-95-96 and 98.5), although one can agree that value 98.5 represents a subtle difference with respect the other three. I would re-write that part with something like “although highly similar between them, PTEN_13 displays a slightly higher…”. In any case I leave the authors to change that section or not.

6. PLOS authors have the option to publish the peer review history of their article (what does this mean?). If published, this will include your full peer review and any attached files.

Reviewer #1: No

Reviewer #2: No

---

## [Author Response · Author response to Decision Letter 0]

8 Dec 2020

Point-by-point replies to editor's comments and Journal Requirements:

Reply: We went through the style and file naming over the entire manuscript in order to match the PLOS One requirements.

Reply: We recognize that we should have been more precise in the text.

This sentence referred to general observations we made based on many different data sets we have accumulated, part of them being not yet published. Compiling all these observations would be difficult, specially since this is not the main topic of this manuscript.

Therefore, in order to support the remark previously formulated as "data not shown", we now selected several examples extracted from only one of our publications (Gogl et al. (2019), Rewiring of RSK-PDZ interactome by linear motif phosphorylation, J. Mol. Biol.; doi: 10.1016/j.jmb.2019.01.038) that is clearly cited.

Below is a table extracted from the supplemental materials available online in Gogl et al. 2019.

PEPTIDES PDZ domains BI mean BI STD # replicates

RSK1-P APBA1-2 -0.10 0.05 2

RSK1-P SYNPO2L -0.10 0.04 2

RSK1-P GRIP2-7 -0.13 0.07 2

RSK1-P APBA3-2 -0.14 0.02 2

RSK1-P MPP4 -0.16 0.02 2

RSK1 MAGI2-1 -0.09 0.06 4

We then substituted the original sentence "In some cases, reproducible negative BI values as low as -0.20 can be observed (data not shown)" in the manuscript by the following one (lines 273-276):

"In some cases, reproducibly significant negative BI values can be observed, for instance for MAGI2-1 binding to RSK1 PBM (BI= -0.09 ± 0.06) and for MPP4-1 or APBA3-2 binding to RSK1 phospho-PBM (BI= -0.16 ± 0.02 and BI= -0.14 ± 0.02) [REF Gogl 2019] as well as for PDZD7-1 binding to HPV16_E6 PBM (BI=-0.18 ± 0.02, a particularly low value in that case) [REF Vincentelli, 2015]."

Those references were already cited earlier in the manuscript.

 

Point-by-point replies to reviewers' comments:

Reviewer #1: The authors combined high-throughput holdup chromatographic assay and competitive fluorescence polarization technique to measure the binding affinity of the PDZ binding motif peptides with a PDZome including 266 PDZ domains. This strategy could be useful to measure other protein interactions. The authors developed a computational processing step to improve the precision of binding intensities. The authors concluded that the mutation at the distal region and acetylation of the core residue could alter the PDZ interaction. These findings make this article meaningful for publication in the journal. After carefully reading, some points which were found from this paper were listed below:

Reply: We thank the reviewer #1 for the careful reading and many insightful comments. We have addressed the points as outlined below. 

1) Line 36-37, “few” means little or none, “many” or “several” instead of "few"could be better here. The words “few” and “a few” were used many times in the paper, please choose proper word to describe the number.

Reply: As requested, we systematically replaced the word 'few' by other words, such as 'limited number', 'some' or 'several' through the entire document.

2) The paragraphs (line 80-103) is to describe the PDZ binding motif, which is not easily understood by not experienced readers. If an extra figure concerning the alignment of the four PTEN peptides, highlight of the key residues, amino acid residues position, consensus sequence, were included in the paper, it would be much helpful.

Reply: As suggested, we add a new figure 1 containing two panels in the introduction: (A) describes the main characteristics discriminating the 3 PDZ classes, while (B) corresponds to a sequence alignment of the PBM constructs used in the present study and is cited in the paragraph "Peptide synthesis" in the M&M section. This implied a renumbering of every figure.

A copy of that figure is shown below.

Fig 1. Summary of the PBM sequences studied in this work.

(A) A classification has been proposed for the PDZ domains according to the sequence consensus observed for the bound PBMs, and is shown here. See text for details. (B) The sequences of the three 11-mer and the 13-mer PBM peptides used in this work are represented. The 11-mer and 13-mer wild-type sequences correspond to the PDZ binding motif of the canonical isoform I of human PTEN (Uniprot ID: P60484), encompassing residues 393 – 403 or 391 – 403, respectively. The mutated residues as compared to wild-type are indicated in bold in peptide sequences. acK: acetylated lysine.

3) Line 92, PTM should be listed in abbreviation.

Reply: Done (line 61).

In the abbreviation list, we also included PTEN, which was originally missing (line 60).

4) The paragraphs (line 124-159) could be shorter. Some contents are duplicated with those in the method and result sections.

Reply: The lines 124 – 159 contained actually 2 paragraphs: 124 – 143 and 145 – 159. The former one corresponds to the very last paragraph of the introduction, summarizing what we are presenting in the article, which implies limited yet unavoidable redundancies with other parts of the text. We have thus reasoned that reviewer #1 actually meant the preceding paragraph, encompassing lines 124 – 143.

There, we acknowledge that some elements of description of teh holdup assay might be redundant between the introduction and the M&M section. However, in our experience, readers (and colleagues) often fail to capture the principle differences between holdup and pulldown approaches. This obliges us to insist on the specificities of the holdup assay, in particular the fact that it focuses on the depletion of the molecule of interest in the flow-through (thereby avoiding washing steps), rather than on its presence on the beads (which require washing steps). Altogether, we slightly modified the M&M section (original lines 193 – 205), in order to both reduce the redundancies and to provide more technical information not presented earlier.

As concerns a duplicated content with the Results section: after careful re-reading we have not detected such redundancies in this section, which is more dedicated to show how we solved technical issues in data quality and reliability rather than the holdup principle itself. In the Result section, we introduced the global quality score, we used the orthogonal unbiased FP technique, an generated not only BI profiles but also log(KD,M) profiles, all points not introduced earlier neither in the introduction, nor in the M&M.

5) Please describe the plasmids, pETG41A and pETG20A (line 167)

Reply: pETG41A and pETG20A are common plasmids allowing to insert the MBP or the THR tag, respectively, in the 5'-end of the gene of interest. See point 6).

6) Please describe MBP and TRX (line 168) and add them to the Abbreviation

Reply: Actually, MBP and TRX were already present in the abbreviation list (lines 57, 62).

From both points 5) and 6) raised by the reviewer #1, we understood that the names of these tags frequently utilized for faciliting protein overexpression are not necessary familiar to all readers. We therefore slightly modified the text in the paragraph "Protein Expression and Purification" in the M&M section in order to better explain the use of those tags and their presence in the two overexpression plasmids (lines 170-171).

7) Line 180: What do the bold letters mean? should be explained!

Reply: The bold letters indicate the modifications introduced as compared to the 11-mer wild-type sequence. By questioning the meaning of the bold letters, the reviewer raised the point in distinguishing between the mutations as it is for K/R in p-1 position, and the insertion of two residues as it is in PTEN_13 as compared to PTEN_11.

The new figure 1 introduced earlier in the manuscript allows to delete the 4 sequences originally presented in this M&M paragraph. We hope the caption of Figure 1 is now properly addressing the two cases, mutation vs. insertion. In particular, a sentence has been added in the legend of Fig 1 to explicitly indicate the meaning of the bold characters that we now only kept for the mutations, K/R and K/AcK (line 92) "The mutated residues as compared to wild-type are indicated in bold in the peptide sequences." 

8) The source of the peptide sequences should be described. The figure mentioned in the point 2 could be included in this section (line 179-190)

Reply: We recognize that this point was not sufficiently described in the original text. The following sentence has been added in the legend of the new figure 1 presented in the introduction of the manuscript (lines 89-91). "The 11-mer and 13-mer peptide sequences correspond to the PDZ binding motif of the canonical isoform I of human PTEN (Uniprot ID: P60484), encompassing residues 393 – 403 or 391 – 403, respectively."

9) Please explain the three PDZ domains in the main text (line 323)

Reply: We are not sure if the reviewer #1 was expecting us to only cite the names of PDZ in the main text rather than just in the caption, or to give additional "biological" information on those 3 PDZ.

We are now citing the names of those 3 PDZ domains in the main text as recommended (line 330). However, we did not provide additional details about those three PDZ domains because they are only used for the purpose of illustrating how the holdup assay can help visualizing strong affinity, weak affinity, or no binding event.

10) This paragraph (line 358-360) should be mentioned in the section of the method

Reply: In this remark, the reviewer #1 is mentioning the sentence by which we explained that we applied the holdup assay to generate the 4 PDZome-binding profiles. It's true that such a sentence might be better suited for the M&M section. However, we still consider this as a part of the result. Indeed, in the result section, we first describe in the previous paragaph the experimental strategy that we set up in order to acquire large numbers of reliable affinity data. Then, we applied this method, starting with the holdup assay, to the 4 PTEN constructs that we are introducing for the first time using this sentence in the Result section. The next part of this paragraph is presenting the statistics and the quality of the holdup data, which were never discussed earlier. We thus think this sentence as essential at the beginning of the paragraph "Generating PDZome-binding profiles of the four PTEN variant PBMs by holdup assay".

11) Line 438, this sentence was not precisely described.

Reply: The original sentence has been split into two sentences, with the hope to fully clarify the description (line 456).

12) Line 548 -549, this sentence is not complete, could cause misleading.

Reply: This sentence has been re-organized, following the classical subject-verb-object order (line 567).

13) Line 568-571, is the two extra residues at the C-terminus of PTEN_13 from the wild type sequence of PTEN or mutation? It should be described in the text. If mutation, 13mer PBM instead of PTEN_11 should be used as a reference.

Reply: The reviewer #1 is absolutely right about this point.

PTEN_13 corresponds to the wild-type sequence of PTEN.

We hope that the insertion of the new figure 1 earlier in the manuscript will address this point, avoiding definitely any confusion.

14) Line 613: This sentence is not precise.

Reply: This sentence has been re-organized, following the classical subject-verb-object order (line 631).

15) Line 621: The acetylation not only increase, but also reduce the affinity of PTEN for PDZ domains (See Fig. 7, Table 1).

Reply:

Below is a scatter plot of the log(KD,M) values observed for PTEN_11 vs. PTEN_Ac, generated with data from the S1 File (unchanged as compared to the originally submitted version).

As we can see, the presence of the acetylation increases most of the affinities. Yet, reviewer #1 is right in saying that this is not true for ALL PDZ domains. We therefore modified the sentence as follows (lines 643-644):

"…, this implies that, for a majority of PDZ domains, their binding affinity to PTEN was reinforced by acetylation."

One might think that this figure might be easier in order to catch the shift toward PTEN_Ac. However, the original figure 7A/ (now figure 8A/) presents the advantage that it allows to directly compare the 3 distinct data sets. Otherwise, we would have needed to show three pairwise scatter plots.

16) Line 699, not always increased, sometimes lost binding affinity (See Fig. 7, table 1)

Reply: If we understood well, it seems to be a confusion for reviewer #1 between the number of targeted PDZ domains and the affinity. We understand that such a confusion can happen, since the discussion right before in the manuscript was mainly talking about affinities and specificities. However, in the original line 699, we are not talking about an increase of the affinities, but an increase of the number of PDZ domains that detectably bound to the PBM. Thus, to avoid any confusion by other readers, we now explicitly added at line 723 a "shift toward stronger affinities although some exceptions have also been observed".

17) In Fig.6., the color scale is not so clear, difficult to be compared. If one column with exact number for -log(KD,(M), it would be much better.

Reply: The suggestion of presenting the numerical data in one additional column per panel is interesting. However, one column could not be enough because several proteins comprise more than one PDZ domain that detectably bind with BI > 0.20. We thus feel that the insertion of 2, if not 3 columns, per panel would become difficult to read.

Furthermore, the main idea in this figure is to show the distribution of PDZ domains with significant KD over the entire protein sequences, getting a global view of the change. We thus prefer to keep the figure as it was originally.

However, we agree with the reviewer #1 than it is important for any interested reader to have access to the precise log(KD,M) values somewhere. This is one of the goals of the S1 File, which combines all numerical values. We also included in the caption of original Figure 6 (now Fig 7) at line 536 the following sentence: " Numerical BI and -log(KD) values can be found in the S1 file."

Reviewer #2: This work from Jané et al. deals with a high-throughput assay aiming the specificity profiling of the whole set of the human PDZ domains versus four variants of the C-terminal region of PTEN protein. Results confirmed that the acetylation of a lysine residue (a PTM that seems to occur in vivo with functional relevance) significantly alters the binding profile. The manuscript is well written, and it encompasses a nicely performed and vast amount of work. I think that it can be published in Plos One with just minor points to be revised. Concretely:

Reply: We warmly thank reviewer #2 for these positive comments. We hope that our results will be useful to other laboratories interested in PDZ/PBM interactions and more generally in PTM impacts.

1. There is a literature reference that should be addressed (Tonikian, R. et al. A Specificity Map for the PDZ Domain Family. PLoS Biology 2008, 6, e239), where a specificity map of PDZ domains is offered. The reference should be included both in the introduction (lines 88-91) and especially in lines 612-616, which are located in the discussion. In Toninkian’s paper, position -1 is also relevant in specificity binding to PDZ domains, and it should be at least commented.

Reply: This is absolutely right. We thank reviewer #2 for reminding us to cite this very relevant work. We thus included the aforementioned reference in the introduction (line 82, new reference #12) and discussed the relevance of this p 1 position in binding specificity as observed by Tonikian et al. from the PBM's point of view (lines 634-638).

2. Having checked reference #31, I am not sure whether the cell lysates are purified based on a His-tagged chromatography or another method or if the lysates are directly used. In the latter case, it seems to me strange to use direct cell lysates. I would like this to be clarified.

Reply: We do work here with cell lysates, not with purified proteins. To clarify this point, we slightly modified the part describing the use of the cell lysate, by including these words (lines 206-207): " cell lysates diluted so that the concentration of the tag-PDZ present in the crude extract is adjusted at 4 µM."

We hope this change will allow to avoid any confusion to the reader.

3. In lines 483-489, there should be a nuance in the comparison of the specificity indexes of the four peptide variants. The percentages are indeed very similar between them (96-95-96 and 98.5), although one can agree that value 98.5 represents a subtle difference with respect the other three. I would re-write that part with something like “although highly similar between them, PTEN_13 displays a slightly higher…”. In any case I leave the authors to change that section or not.

Reply: We agree with the reviewer #2 that we should be more careful with our conclusions and have re-written this paragraph (lines 500 – 505):

"Although the index values are highly similar between PTEN_11, PTEN_Ac, and PTEN_KR [(95.8 ± 2.3)%, (94.9 ± 1.5)%, (95.9 ± 2.1)%, respectively], the extended wild-type peptide PTEN_13 displays a slightly higher specificity index [(98.5 ± 0.4)%; p-value = 0.039 for PTEN_Ac vs. PTEN_13 considering a Fisher's exact test] which could be indicative of a higher specificity towards several selected PDZ domains."

---

## [Editor Report · Decision Letter 1]

14 Dec 2020

Interactomic affinity profiling by holdup assay: acetylation and distal residues impact the PDZome-binding specificity of PTEN phosphatase

PONE-D-20-29437R1

Dear Dr. Nominé,

We’re pleased to inform you that your manuscript has been judged scientifically suitable for publication and will be formally accepted for publication once it meets all outstanding technical requirements.

Kind regards,

Petri Kursula

Academic Editor

PLOS ONE
---

## [Editor Report · Acceptance letter]

18 Dec 2020

PONE-D-20-29437R1 

Interactomic affinity profiling by holdup assay: acetylation and distal residues impact the PDZome-binding specificity of PTEN phosphatase 

Dear Dr. Nominé:

I'm pleased to inform you that your manuscript has been deemed suitable for publication in PLOS ONE. Congratulations! Your manuscript is now with our production department. 

Kind regards, 

on behalf of

Prof. Petri Kursula 

Academic Editor

PLOS ONE